# PROBABILISTIC PLANNING WITH SEQUENTIAL MONTE CARLO METHODS

**Alexandre Piché**[*12]**, Valentin Thomas**[*12]**, Cyril Ibrahim** [2]**, Yoshua Bengio** [13]**, Chris Pal** [1245]

[1] Mila, Université de Montréal

[2] Element AI

[3] CIFAR Senior Fellow

[4] Mila, Polytechnique Montréal

[5] Canada CIFAR AI Chair

## ABSTRACT

In this work, we propose a novel formulation of planning which views it as a probabilistic inference problem over future optimal trajectories. This enables us to use sampling methods, and thus, tackle planning in continuous domains using a fixed computational budget. We design a new algorithm, Sequential Monte Carlo Planning, by leveraging classical methods in Sequential Monte Carlo and Bayesian smoothing in the context of *control as inference*. Furthermore, we show that Sequential Monte Carlo Planning can capture multimodal policies and can quickly learn continuous control tasks.

## 1 INTRODUCTION

To exhibit intelligent behaviour machine learning agents must be able to learn quickly, predict the consequences of their actions, and explain how they will react in a given situation. These abilities are best achieved when the agent efficiently uses a model of the world to plan future actions. To date, planning algorithms have yielded very impressive results. For instance, Alpha Go (Silver et al., 2017) relied on Monte Carlo Tree Search (MCTS) (Kearns et al., 2002) to achieve super human performances. Cross entropy methods (CEM) (Rubinstein & Kroese, 2004) have enabled robots to perform complex nonprehensile manipulations (Finn & Levine, 2017) and algorithms to play successfully Tetris (Szita & Lörincz, 2006). In addition, iterative linear quadratic regulator (iLQR) (Kalman et al., 1960; Kalman, 1964; Todorov & Li, 2005) enabled humanoid robots tasks to get up from an arbitrary seated pose (Tassa et al., 2012).

Despite these successes, these algorithms make strong underlying assumptions about the environment. First, MCTS requires a discrete setting, limiting most of its successes to discrete games with known dynamics. Second, CEM assumes the distribution over future trajectories to be Gaussian, i.e. unimodal. Third, iLQR assumes that the dynamics are locally linear-Gaussian, which is a strong assumption on the dynamics and would also assume the distribution over future optimal trajectories to be Gaussian. For these reasons, planning remains an open problem in environments with continuous actions and complex dynamics. In this paper, we address the limitations of the aforementioned planning algorithms by creating a more general view of planning that can leverage advances in deep learning (DL) and probabilistic inference methods. This allows us to approximate arbitrary complicated distributions over trajectories with non-linear dynamics.

We frame planning as density estimation problem over optimal future trajectories in the context of *control as inference* (Dayan & Hinton, 1997; Toussaint & Storkey, 2006; Toussaint, 2009; Rawlik et al., 2010; 2012; Ziebart, 2010; Levine & Koltun, 2013). This perspective allows us to make use of tools from the inference research community and, as previously mentioned, model any distribution over future trajectories. The planning distribution is complex since trajectories consist of an intertwined sequence of states and actions. Sequential Monte Carlo (SMC) (Stewart & McCarty, 1992; Gordon et al., 1993; Kitagawa, 1996) methods are flexible and efficient to model such a

---

*both authors contributed equally.

distribution by sequentially drawing from a simpler proposal distribution. From the SMC perspective, the policy can be seen as the proposal and a learned model of the world as the propagation distribution. This provides a natural way to combine model-free and model-based RL.

**Contribution.** We depict the problem of planning as one of density estimation that can be estimated using SMC methods. We introduce a novel planning strategy based on the SMC class of algorithms, in which we treat the policy as the proposed distribution to be learned. We investigate how our method empirically compares with existing model-based methods and a strong model-free baseline on the standard benchmark Mujoco (Todorov et al., 2012).

## 2 BACKGROUND

### 2.1 CONTROL AS INFERENCE

We consider the general case of a Markov Decision Process (MDP) $\{\mathcal{S}, \mathcal{A}, p_{\text{env}}, r, \gamma, \mu\}$ where $\mathcal{S}$ and $\mathcal{A}$ represent the state and action spaces respectively. We use the letters $\mathbf{s}$ and $\mathbf{a}$ to denote states and actions, which we consider to be continuous vectors. Further notations include: $p_{\text{env}}(\mathbf{s}'|\mathbf{s}, \mathbf{a})$ as the state transition probability of the environment, $r(\mathbf{s}, \mathbf{a})$ as the reward function, and $\gamma \in [0, 1)$ as the discount factor. $\mu$ denotes the probability distribution over initial states.

This work focuses on an episodic formulation, with a fixed end-time of $T$. We define a trajectory as a sequence of state-action pairs $\mathbf{x}_{t:T} = \{(\mathbf{s}_t, \mathbf{a}_t), \ldots, (\mathbf{s}_T, \mathbf{a}_T)\}$, and we use the notation $\pi$ for a policy which represents a distribution over actions conditioned on a state. Here $\pi$ is parametrized by a neural network with parameters $\theta$. The notation $q_\theta(\mathbf{x}_{1:T}) = \mu(\mathbf{s}_1) \prod_{t \geq 1}^{T-1} p_{\text{env}}(\mathbf{s}_{t+1}|\mathbf{s}_t, \mathbf{a}_t) \prod_{t \geq 1}^{T} \pi_\theta(\mathbf{a}_t|\mathbf{s}_t)$ denotes the probability of a trajectory $\mathbf{x}_{1:T}$ under policy $\pi_\theta$.

Traditionally, in reinforcement learning (RL) problems, the goal is to find the optimal policy that maximizes the expected return $\mathbb{E}_{q_\theta}[\sum_{t=1}^{T} \gamma^t r_t]$. However, it is useful to frame RL as an inference problem within a probabilistic graphical framework (Rawlik et al., 2012; Toussaint & Storkey, 2006; Levine, 2018). First, we introduce an auxiliary binary random variable $\mathcal{O}_t$ denoting the "optimality" of a pair $(\mathbf{s}_t, \mathbf{a}_t)$ at time $t$ and define its probability[1] as $p(\mathcal{O}_t = 1|\mathbf{s}_t, \mathbf{a}_t) = \exp(r(\mathbf{s}_t, \mathbf{a}_t))$. $\mathcal{O}$ is a convenience variable only here for the sake of modeling. By considering the variables $(\mathbf{s}_t, \mathbf{a}_t)$ as latent and $\mathcal{O}_t$ as observed, we can construct a Hidden Markov Model (HMM) as depicted in figure 2.1. Notice that the link $\mathbf{s} \to \mathbf{a}$ is not

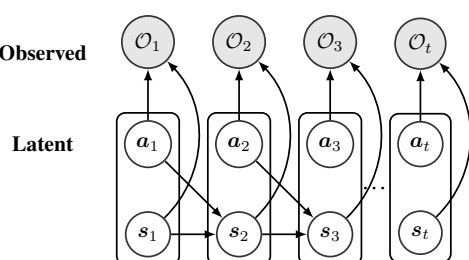

Figure 2.1: $\mathcal{O}_t$ is an observed *optimality* variable with probability $p(\mathcal{O}_t|\mathbf{s}_t, \mathbf{a}_t) = \exp(r(\mathbf{s}_t, \mathbf{a}_t))$. $\mathbf{x}_t = (\mathbf{s}_t, \mathbf{a}_t)$ are the state-action pair variables considered here as latent.

present in figure 2.1 as the dependency of the optimal action on the state depends on the future observations. In this graphical model, the optimal policy is expressed as $p(\mathbf{a}_t|\mathbf{s}_t, \mathcal{O}_{t:T})$.

The posterior probability of this graphical model can be written as[2]:

$$p(\mathbf{x}_{1:T}|\mathcal{O}_{1:T}) \propto p(\mathbf{x}_{1:T}, \mathcal{O}_{1:T}) = \mu(\mathbf{s}_1) \prod_{t=1}^{T-1} p_{\text{env}}(\mathbf{s}_{t+1}|\mathbf{a}_t, \mathbf{s}_t) \exp\big(\sum_{t=1}^{T} r(\mathbf{s}_t, \mathbf{a}_t) + \log p(\mathbf{a}_t)\big).$$
(2.1)

It appears clearly that finding optimal trajectories is equivalent to finding plausible trajectories yielding a high return.

---

[1] as in Levine (2018), if the rewards are bounded above, we can always remove a constant so that the probability is well defined.

[2] Notice that in the rest of the paper, we will abusively remove the product of the action priors $\prod_{t=1}^{T} p(\mathbf{a}_t) = \exp\big(\sum_{t=1}^{T} \log p(\mathbf{a}_t)\big)$ from the joint as in Levine (2018). We typically consider this term either constant or already included in the reward function. See Appendix A.2 for details.

Many *control as inference* methods can be seen as approximating the density by optimizing its variational lower bound: $\log p(\mathcal{O}_{1:T}) \geq \mathbb{E}_{\mathbf{x}_{1:T} \sim q_\theta}[\sum_{t=1}^{T} r(\mathbf{s}_t, \mathbf{a}_t) - \log \pi_\theta(\mathbf{a}_t|\mathbf{s}_t)]$ (Rawlik et al., 2012; Toussaint, 2009). Instead of directly differentiating the variational lower bound for the whole trajectory, it is possible to take a message passing approach such as the one used in Soft Actor-Critic (SAC) (Haarnoja et al., 2018) and directly estimate the optimal policy $p(\mathbf{a}_t|\mathbf{s}_t, \mathcal{O}_{t:T})$ using the backward message, i.e a soft $Q$ function instead of the Monte Carlo return.

## 2.2 Sequential Monte Carlo methods

Since distributions over trajectories are complex, it is often difficult or impossible to directly draw samples from them. Fortunately in statistics, there are successful strategies for drawing samples from complex sequential distributions, such as SMC methods.

For simplicity, in the remainder of this section we will overload the notation and refer to the target distribution as $p(\mathbf{x})$ and the proposal distribution as $q(\mathbf{x})$. We wish to draw samples from $p$ but we only know its unnormalized density. We will use the proposal $q$ to draw samples and estimate $p$. In the next section, we will define the distributions $p$ and $q$ in the context of planning.

**Importance sampling (IS):**    When $\mathbf{x}$ can be efficiently sampled from another simpler distribution $q$ i.e. the proposal distribution, we can estimate the likelihood of any point $\mathbf{x}$ under $p$ straightforwardly by computing the *unnormalized importance sampling weights* $w(\mathbf{x}) \propto \frac{p(\mathbf{x})}{q(\mathbf{x})}$ and using the identity $p(\mathbf{x}) = \bar{w}(\mathbf{x})q(\mathbf{x})$ where $\bar{w}(\mathbf{x}) = \frac{w(\mathbf{x})}{\int w(\mathbf{x})q(\mathbf{x})d\mathbf{x}}$ is defined as the *normalized importance sampling weights*. In practice, one draws $N$ samples from $q$: $\{\mathbf{x}^{(n)}\}_{n=1}^{N} \sim q$; these are referred to as *particles*. The set of particles $\{\mathbf{x}^{(n)}\}_{n=1}^{N}$ associated with their weights $\{w^{(n)}\}_{n=1}^{N}$ are simulations of samples from $p$. That is, we approximate the density $p$ with a weighted sum of diracs from samples of $q$:

$$p(\mathbf{x}) \approx \sum_{n=1}^{N} \bar{w}^{(n)} \delta_{\mathbf{x}^{(n)}}(\mathbf{x}), \text{ with } \mathbf{x}^{(n)} \text{ sampled from } q$$

where $\delta_{\mathbf{x}_0}(\mathbf{x})$ denotes the Dirac delta mass located as $\mathbf{x}_0$.

**Sequential Importance Sampling (SIS):**    When our problem is sequential in nature $\mathbf{x} = \mathbf{x}_{1:T}$, sampling $\mathbf{x}_{1:T}$ at once can be a challenging or even intractable task. By exploiting the sequential structure, the unnormalized weights can be updated iteratively in an efficient manner: $w_t(\mathbf{x}_{1:t}) = w_{t-1}(\mathbf{x}_{1:t-1})\frac{p(\mathbf{x}_t|\mathbf{x}_{1:t-1})}{q(\mathbf{x}_t|\mathbf{x}_{1:t-1})}$. We call this the **update step**. This enables us to sample sequentially $\mathbf{x}_t \sim q(\mathbf{x}_t|\mathbf{x}_{1:t-1})$ to finally obtain the set of particles $\{\mathbf{x}_{1:T}^{(n)}\}$ and their weights $\{w_T^{(n)}\}$ linearly in the horizon $T$.

**Sequential Importance Resampling (SIR):**    When the horizon $T$ is long, samples from $q$ usually have a low likelihood under $p$, and thus the quality of our approximation decreases exponentially with $T$. More concretely, the unnormalized weights $w_t^{(n)}$ converge to 0 with $t \to \infty$. This usually causes the normalized weight distribution to degenerate, with one weight having a mass of 1 and the others a mass of 0. This phenomenon is known as *weight impoverishment*.

One way to address weight impoverishment is to add a **resampling step** where each particle is stochastically resampled to higher likelihood regions at each time step. This can typically reduce the variance of the estimation from growing *exponentially* with $t$ to growing *linearly*.

## 3 Planning as probabilistic inference

In the context of *control as inference*, it is natural to see planning as the act of approximating a distribution of optimal future trajectories via simulation. In order to plan, an agent must possess a model of the world that can accurately capture consequences of its actions. In cases where multiple trajectories have the potential of being optimal, the agent must rationally partition its computational resources to explore each possibility. Given finite time, the agent must limit its planning to a finite horizon $h$. We, therefore, define *planning* as the act of approximating the optimal distribution over

trajectories of length $h$. In the control-as-inference framework, this distribution is naturally expressed as $p(\mathbf{a}_1, \mathbf{s}_2, \ldots \mathbf{s}_h, \mathbf{a}_h | \mathcal{O}_{1:T}, \mathbf{s}_1)$, where $\mathbf{s}_1$ represents our current state.

## 3.1 PLANNING AND BAYESIAN SMOOTHING

As we consider the current state $\mathbf{s}_1$ given, it is equivalent and convenient to focus on the planning distribution with horizon $h$: $p(\mathbf{x}_{1:h} | \mathcal{O}_{1:T})$. Bayesian smoothing is an approach to the problem of estimating the distribution of a latent variable conditioned on all past and future observations. One method to perform smoothing is to decompose the posterior with the *two-filter formula* (Bresler, 1986; Kitagawa, 1994):

$$p(\mathbf{x}_{1:h} | \mathcal{O}_{1:T}) \propto p(\mathbf{x}_{1:h} | \mathcal{O}_{1:h}) \cdot p(\mathcal{O}_{h+1:T} | \mathbf{x}_h) \tag{3.1}$$

This corresponds to a forward-backward messages factorization in a Hidden Markov Model as depicted in figure 3.1. We broadly underline in orange forward variables and in blue backward variables in the rest of this section.

**Filtering** is the task of estimating $p(\mathbf{x}_{1:t} | \mathcal{O}_{1:t})$: the probability of a latent variable conditioned on all past observations. In contrast, **smoothing** estimates $p(\mathbf{x}_{1:t} | \mathcal{O}_{1:T})$: the density of a latent variable conditioned on all the past and future measurements.

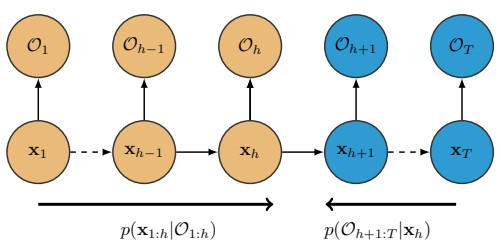

In the belief propagation algorithm for HMMs, these probabilities correspond to the forward message $\alpha_h(\mathbf{x}_h) = p(\mathbf{x}_{1:h} | \mathcal{O}_{1:h})$ and backward message $\beta_h(\mathbf{x}_h) = p(\mathcal{O}_{h+1:T} | \mathbf{x}_h)$, both of which are computed recursively. While in discrete spaces these forward and backward messages can be estimated using the sum-product algorithm, its complexity scales with the square of the space dimension making it unsuitable for continuous tasks. We will now devise efficient strategies for estimating reliably the full posterior using the SMC methods covered in section 2.2.

Figure 3.1: Factorization of the HMM into forward (orange) and backward (blue) messages. Estimating the forward message is filtering, estimating the value of the latent knowing all the observations is smoothing.

## 3.2 THE BACKWARD MESSAGE AND THE VALUE FUNCTION

The backward message $p(\mathcal{O}_{h+1:T} | \mathbf{x}_h)$ can be understood as the answer to: "What is the probability of following an optimal trajectory from the next time step on until the end of the episode, given my current state?". Importantly, this term is closely related to the notion of *value function* in RL. Indeed, in the control-as-inference framework, the state- and action-value functions are defined as $V(\mathbf{s}_h) \triangleq \log p(\mathcal{O}_{h:T} | \mathbf{s}_h)$ and $Q(\mathbf{s}_h, \mathbf{a}_h) \triangleq \log p(\mathcal{O}_{h:T} | \mathbf{s}_h, \mathbf{a}_h)$ respectively. They are solutions of a soft-Bellman equation that differs a little from the traditional Bellman equation (O'Donoghue et al., 2016; Nachum et al., 2017; Schulman et al., 2017; Abdolmaleki et al., 2018). A more in depth explanation can be found in (Levine, 2018). We can show subsequently that:

$$p(\mathcal{O}_{h+1:T} | \mathbf{x}_h) = \mathbb{E}_{\mathbf{s}_{h+1} | \mathbf{x}_h} \left[ \exp\left( V(\mathbf{s}_{h+1}) \right) \right] \tag{3.2}$$

Full details can be found in Appendix A.3. Estimating the backward message is then equivalent to learning a value function. This value function as defined here is the same one used in Maximum Entropy RL (Ziebart, 2010).

## 3.3 SEQUENTIAL WEIGHT UPDATE

Using the results of the previous subsections we can now derive the full update of the sequential importance sampling weights. To be consistent with the terminology of section 2.2, we call $p(\mathbf{x}_{1:h} | \mathcal{O}_{1:T})$

the target distribution and $q_\theta(\mathbf{x}_{1:h})$ the proposal distribution. The sequential weight update formula is in our case:

$$
\begin{aligned}
w_t &= w_{t-1} \cdot \frac{p(\mathbf{x}_t|\mathbf{x}_{1:t-1}, \mathcal{O}_{1:T})}{q_\theta(\mathbf{x}_t|\mathbf{x}_{1:t-1})} \\
&\propto w_{t-1} \frac{1}{q_\theta(\mathbf{x}_t|\mathbf{x}_{1:t-1})} \frac{p(\mathbf{x}_{1:t}|\mathcal{O}_{1:t})}{p(\mathbf{x}_{1:t-1}|\mathcal{O}_{1:t-1})} \frac{p(\mathcal{O}_{t+1:T}|\mathbf{x}_t)}{p(\mathcal{O}_{t:T}|\mathbf{x}_{t-1})} \\
&\propto w_{t-1} \cdot \mathbb{E}_{\mathbf{s}_{t+1}|\mathbf{s}_t, \mathbf{a}_t}[\exp\left(A(\mathbf{s}_t, \mathbf{a}_t, \mathbf{s}_{t+1})\right)]
\end{aligned}
$$

Where

$$
A(\mathbf{s}_t, \mathbf{a}_t, \mathbf{s}_{t+1}) = r_t - \log \pi_\theta(\mathbf{a}_t|\mathbf{s}_t) + V(\mathbf{s}_{t+1}) - \log \mathbb{E}_{\mathbf{s}_t|\mathbf{s}_{t-1}, \mathbf{a}_{t-1}}[\exp\left(V(\mathbf{s}_t)\right)] \tag{3.3}
$$

is akin to a maximum entropy advantage function. The change in weight can be interpreted as sequentially correcting our expectation of the return of a trajectory.

The full derivation is available in Appendix A.4. Our algorithm is similar to the Auxilliary Particle Filter (Pitt & Shephard, 1999) which uses a one look ahead simulation step to update the weights. Note that we have assumed that our model of the environment was perfect to obtain this slightly simplified form. This assumption is made by most planning algorithms (LQR, CEM ...): it entails that our plan is only as good as our model is. A typical way to mitigate this issue and be more robust to model errors is to re-plan at each time step; this technique is called Model Predictive Control (MPC) and is commonplace in control theory.

## 3.4 SEQUENTIAL MONTE CARLO PLANNING ALGORITHM

We can now use the computations of previous subsections to derive the full algorithm. We consider the root state of the planning to be the current state $\mathbf{s}_t$. We aim at building a set of particles $\{\mathbf{x}_{t:t+h}^{(n)}\}_{n=1}^N$ and their weights $\{w_{t+h}^{(n)}\}_{n=1}^N$ representative of the planning density $p(\mathbf{x}_{t:t+h}|\mathcal{O}_{1:T})$ over optimal trajectories. We use SAC (Haarnoja et al., 2018) for the policy and value function, but any other Maximum Entropy policy can be used for the proposal distribution. Note that we used the value function estimated by SAC as a proxy the optimal one as it is usually done by actor critic methods.

We summarize the proposed algorithm in Algorithm 1. At each step, we sample from the proposal distribution or model-free agent (**line 6**) and use our learned model to sample the next state and reward (**line 7**). We then update the weights (**line 8**). In practice we only use one sample to estimate the expectations, thus we may incur a small bias. The resampling step is then performed (**line 10-11**) by resampling the trajectories according to their weight. After the planning horizon is reached, we sample one of our trajectories (**line 13**) and execute its first action into the environment (**line 15-16**). The observations $(\mathbf{s}_t, \mathbf{a}_t, r_t, \mathbf{s}_{t+1})$ are then collected and added to a buffer (**line 17**) used to train the model as well as the policy and value function of the model-free agent. An alternative algorithm that does not use the resampling step (SIS) is highlighted in Algorithm 2 in Appendix A.6.

---

**Algorithm 1** SMC Planning using SIR

1: **for** $t$ in $\{1, \ldots, T\}$ **do**
2: $\quad \{\mathbf{s}_t^{(n)} = \mathbf{s}_t\}_{n=1}^N$
3: $\quad \{w_t^{(n)} = 1\}_{n=1}^N$
4: $\quad$ **for** $i$ in $\{t, \ldots, t+h\}$ **do**
5: $\qquad$ // Update
6: $\qquad \{\mathbf{a}_i^{(n)} \sim \pi(\mathbf{a}_i^{(n)}|\mathbf{s}_i^{(n)})\}_{n=1}^N$
7: $\qquad \{\mathbf{s}_{i+1}^{(n)}, r_i^{(n)} \sim p_{\text{model}}(\cdot|\mathbf{s}_i^{(n)}, \mathbf{a}_i^{(n)})\}_{n=1}^N$
8: $\qquad \{w_i^{(n)} \propto w_{i-1}^{(n)} \cdot \exp\left(A(\mathbf{s}_i^{(n)}, \mathbf{a}_i^{(n)}, \mathbf{s}_{i+1}^{(n)})\right)\}_{n=1}^N$
9: $\qquad$ // Resampling
10: $\qquad \{\mathbf{x}_{1:i}^{(n)}\}_{n=1}^N \sim \text{Mult}(n; w_i^{(1)}, \ldots, w_i^{(N)})$
11: $\qquad \{w_i^{(n)} = 1\}_{n=1}^N$
12: $\quad$ **end for**
13: $\quad$ Sample $n \sim \text{Uniform}(1, N)$.
14: $\quad$ // Model Predictive Control
15: $\quad$ Select $\mathbf{a}_t$, first action of $\mathbf{x}_{t:t+h}^{(n)}$
16: $\quad \mathbf{s}_{t+1}, r_t \sim p_{\text{env}}(\cdot|\mathbf{s}_t, \mathbf{a}_t)$
17: $\quad$ Add $(\mathbf{s}_t, \mathbf{a}_t, r_t, \mathbf{s}_{t+1})$ to buffer $\mathcal{B}$
18: $\quad$ Update $\pi$, $V$ and $p_{\text{model}}$ with $\mathcal{B}$
19: **end for**

---

A schematic view of the algorithm can also be found on figure 3.2.

## 3.5 OPTIMISM BIAS AND CONTROL AS INFERENCE

We now discuss shortcomings our approach to planning as inference may suffer from, namely encouraging risk seeking policies.

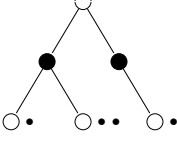 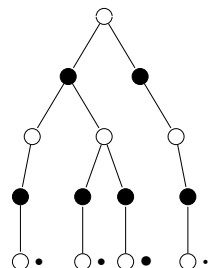 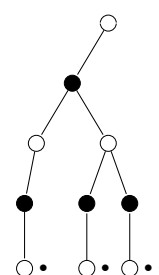

(a) **Configuration at time** $t-1$: we have the root white node $\mathbf{s}_{t-1}$, the actions $\mathbf{a}_{t-1}^{(n)}$ are black nodes and the leaf nodes are the $\mathbf{s}_t^{(n)}$. We have one particle on the leftmost branch, two on the central branch and one on the rightmost branch.

(b) **Update:** New actions and states are sampled from the proposal distribution and model. The particle sizes are proportional to their importance weight $w_t$.

(c) **Resampling:** after sampling with replacement the particles relatively to their weight, the less promising branch was cut while the most promising has now two particles.

Figure 3.2: Schematic view of Sequential Monte Carlo planning. In each tree, the white nodes represent states and black nodes represent actions. Each bullet point near a state represents a particle, meaning that this particle contains the total trajectory of the branch. The root of the tree represents the root planning state, we expand the tree downward when planning.

**Bias in the objective:** Trajectories having a high likelihood under the posterior defined in Equation 2.1 are not necessarily trajectories yielding a high *mean* return. Indeed, as $\log \mathbb{E}_p \big[ \exp R(\mathbf{x}) \big] \geq \mathbb{E}_p \big[ R(\mathbf{x}) \big]$ we can see that the objective function we maximize is an *upper bound* on the quantity of interest: the mean return. This can lead to seeking risky trajectories as one very good outcome in $\log \mathbb{E} \exp$ could dominate all the other potentially very low outcomes, even if they might happen more frequently. This fact is alleviated when the dynamics of the environment are close to deterministic (Levine, 2018). Thus, this bias does not appear to be very detrimental to us in our experiments 4 as our environments are fairly close to deterministic. The bias in the objective also appears in many control as inference works such as Particle Value Functions (Maddison et al., 2017a) and the probabilistic version of LQR proposed in Toussaint (2009).

**Bias in the model:** A distinct but closely related problem arises when one trains jointly the policy $\pi_\theta$ and the model $p_{\text{model}}$, i.e if $q(\mathbf{x}_{1:T})$ is directly trained to approximate $p(\mathbf{x}_{1:T}|\mathcal{O}_{1:T})$. In that case, $p_{\text{model}}(\mathbf{s}_{t+1}|\mathbf{s}_t, \mathbf{a}_t)$ will not approximate $p_{\text{env}}(\mathbf{s}_{t+1}|\mathbf{s}_t, \mathbf{a}_t)$ but $p_{\text{env}}(\mathbf{s}_{t+1}|\mathbf{s}_t, \mathbf{a}_t, \mathcal{O}_{t:T})$ (Levine, 2018). This means the model we learn has an optimism bias and learns transitions that are overly optimistic and do no match the environment's behavior. This issue is simply solved by training the model separately from the policy, on transition data contained in a buffer as seen on line 18 of Algorithm 1.

## 4 EXPERIMENTS

### 4.1 TOY EXAMPLE

In this section, we show how SMCP can deal with multimodal policies when planning. We believe multimodality is useful for exploring since it allows us to keep a distribution over many promising trajectories and also allows us to adapt to changes in the environment e.g. if a path is suddenly blocked.

We applied two version of SMCP: i) with a resampling step (SIR) ii) without a resampling step (SIS) and compare it to CEM on a simple 2D point mass environment 4.1. Here, the agent can control the displacement on $(x, y)$ within the square $[0, 1]^2$, $\mathbf{a} = (\Delta x, \Delta y)$ with maximum magnitude $\|\mathbf{a}\| = 0.05$. The starting position ($\bullet$) of the agent is $(x = 0, y = 0.5)$, while the goal ($\star$) is at $\mathbf{g} = (x = 1, y = 0.5)$. The reward is the agent's relative closeness increment to the goal:

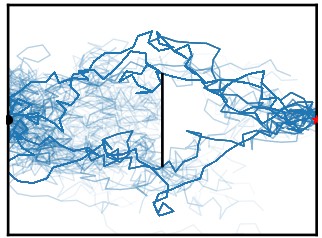

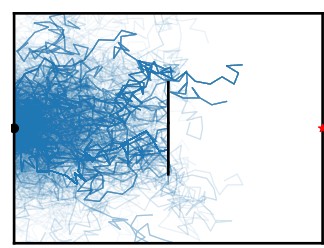

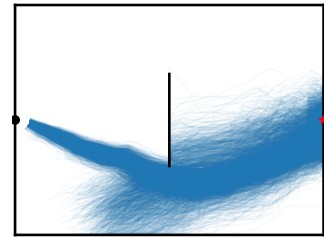

(a) Sequential Importance Resampling (SIR): when resampling the trajectories at each time step, the agent is able to focus on the promising trajectories and does not collapse on a single mode.

(b) Sequential Importance Sampling (SIS): if we do not perform the resampling step the agent spends most of its computation on uninteresting trajectories and was not able to explore as well.

(c) CEM: here the agent samples all the actions at once from a Gaussian with learned mean and covariance. We needed to update the parameters 50 times for the agent to find one solution, but it forgot the other one.

Figure 4.1: Comparison of three methods on the toy environment. The agent ($\bullet$) must go to the goal ($\star$) while avoiding the wall ( $|$ ) in the center. The proposal distribution is taken to be an isotropic gaussian. Here we plot the planning distribution imagined at $t = 0$ for three different agents. A darker shade of blue indicates a higher likelihood of the trajectory. Only the agent using Sequential Importance Resampling was able to find good trajectories while not collapsing on a single mode.

$r_t = 1 - \frac{||\mathbf{s}_{t+1} - \mathbf{g}||^2}{||\mathbf{s}_t - \mathbf{g}||^2}$. However, there is a partial wall at the centre of the square leading to two optimal trajectories, one choosing the path below the wall and one choosing the path above.

The proposal is an isotropic normal distribution for each planning algorithm, and since the environment's dynamics are known, there is no need for learning: the only difference between the three methods is how they handle planning. We also set the value function to $0$ for SIR and SIS as we do not wish to perform any learning. We used $1500$ particles for each method, and updated the parameters of CEM until convergence. Our experiment 4.1 shows how having particles can deal with multimodality and how the resampling step can help to focus on the most promising trajectories.

## 4.2 CONTINUOUS CONTROL BENCHMARK

The experiments were conducted on the Open AI Gym Mujoco benchmark suite (Brockman et al., 2016; Todorov et al., 2012). To understand how planning can increase the learning speed of RL agents we focus on the 250000 first time steps. The Mujoco environments provide a complex benchmark with continuous states and actions that requires exploration in order to achieve state-of-the-art performances.

The environment model used for our planning algorithm is the same as the probabilistic neural network used by Chua et al. (2018), it minimizes a gaussian negative log-likelihood model:

$$\mathcal{L}_{\text{Gauss}}(\theta) = \frac{1}{2} \sum_{n=1}^{N} [\mu_\theta(\mathbf{s}_n, \mathbf{a}_n) - (\mathbf{s}_{n+1} - \mathbf{s}_n)]^\top \Sigma_\theta^{-1}(\mathbf{s}_n, \mathbf{a}_n)[\mu_\theta(\mathbf{s}_n, \mathbf{a}_n) - (\mathbf{s}_{n+1} - \mathbf{s}_n)] + \log \det \Sigma_\theta(\mathbf{s}_n, \mathbf{a}_n),$$

where $\Sigma_\theta$ is diagonal and the transitions $(\mathbf{s}_n, \mathbf{a}_n, \mathbf{s}_{n+1})$ are obtained from the environment. We added more details about the architecture and the hyperparameters in the appendix A.5.

We included two popular planning algorithms on Mujoco as baselines: CEM (Chua et al., 2018) and Random Shooting (RS) (Nagabandi et al., 2017). Furthermore, we included SAC (Haarnoja et al., 2018), a model free RL algorithm, since i) it has currently one of the highest performances on Mujoco tasks, which make it a very strong baseline, and ii) it is a component of our algorithm, as we use it as a proposal distribution in the planning phase.

Our results suggest that SMCP does not learn as fast as CEM and RS initially as it heavily relies on estimating a good value function. However, SMCP quickly achieves higher performances than CEM and RS. SMCP also learns faster than SAC because it was able to leverage information from the model early in training.

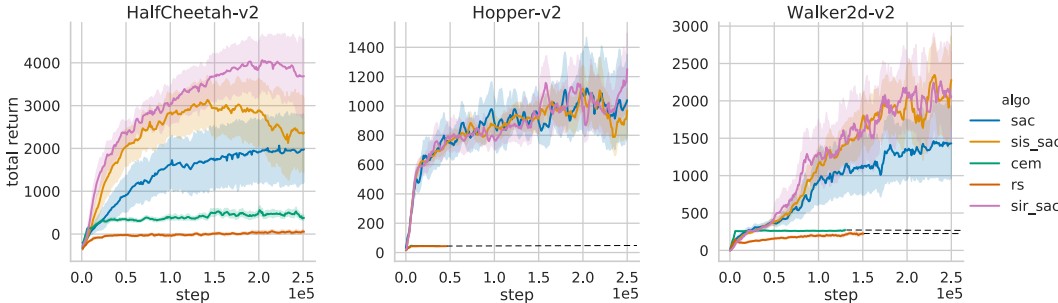

Figure 4.2: Training curves on the Mujoco continuous control benchmarks. Sequential Monte Carlo Planning both with resampling (SIR) (pink) and without (SIS) (orange) learns faster than the Soft Actor-Critic model-free baseline (blue) and achieves higher asymptotic performances than the planning methods (Cross Entropy Methods and Random Shooting). The shaded area represents the standard deviation estimated by bootstrap over 20 seeds as implemented by the Seaborn package.

Note that our results differ slightly from the results usually found in the model-based RL literature. This is because we are tackling a more difficult problem: estimating the transitions and the reward function. We are using unmodified versions of the environments which introduces many hurdles. For instance, the reward function is challenging to learn from the state and very noisy. Usually, the environments are modified such that their reward can be computed directly from the state e.g. Chua et al. (2018) [3].

As in Henderson et al. (2017), we assess the significance of our results by running each algorithm with multiple seeds (20 random seeds in our case, from seed 0 to seed 19) and we perform a statistical significance test following Colas et al. (2018). We test the hypothesis that our mean return on the last 100k steps is higher than the one obtained by SAC. Our results are significant to the 5% for HalfCheetah and Walker2d. See Appendix A.7 for additional details.

We also report some additional experimental results such as effective sample size and model loss in Appendix A.8.

## 5 RELATED WORK

**Planning as inference:** Seeing planning as an inference problem has been explored in cognitive neuroscience by Botvinick & Toussaint (2012) and Solway & Botvinick (2012). While shedding light on how Bayesian inference could be used in animal and human reasoning, it does not lead to a practical algorithm usable in complex environments. In the reinforcement learning literature, we are only aware of Attias (2003) and Toussaint & Goerick (2007) that initially framed planning as an inference problem. However, both works make simplifying assumptions on the dynamics and do not attempt to capture the full posterior distribution.

**Control and planning:** In the control theory literature, particle filters are usually used for inferring the true state of the system which is then used for control (Andrieu et al., 2004). Kantas et al. (2009) also combined SMC and MPC methods. While their algorithm is similar to ours, the distribution they approximate is not the Bayesian posterior, but a distribution which converges to a Dirac on the best trajectory. More recently, Kurutach et al. (2018) achieved promising results on a rope manipulation task using generative adversarial network (Goodfellow et al., 2014) to generate future trajectories.

**Model based RL:** Recent work has been done in order to improve environment modeling and account for different type of uncertainties. Chua et al. (2018) compared the performance of models that account for both aleatoric and epistemic uncertainties by using an ensemble of probabilistic models. Ha & Schmidhuber (2018) combined the variational autoencoder (Kingma & Welling, 2013) and a LSTM (Hochreiter & Schmidhuber, 1997) to model the world. Buckman et al. (2018) used a model to improve the target for temporal difference (TD) learning. Note that this line of work is

---

[3]https://github.com/kchua/handful-of-trials/tree/e1a62f217508a384e49ecf7d16a3249e187bcff9/dmbrl/env

complementary to ours as SMCP could make use of such models. Other works have been conducted in order to directly learn how to use a model (Guez et al., 2018; Weber et al., 2017; Buesing et al., 2018).

**Particle methods and variational inference:** Gu et al. (2015) learn a good proposal distribution for SMC methods by minimizing the KL divergence with the optimal proposal. It is conceptually similar to the way we use SAC (Haarnoja et al., 2018) but it instead minimizes the reverse KL to the optimal proposal. Further works have combined SMC methods and variational inference (Naesseth et al., 2017; Maddison et al., 2017b; Le et al., 2017) to obtain lower variance estimates of the distribution of interest.

## 6 CONCLUSION AND FUTURE WORK

In this work, we have introduced a connection between planning and inference and showed how we can exploit advances in deep learning and probabilistic inference to design a new efficient and theoretically grounded planning algorithm. We additionally proposed a natural way to combine model-free and model-based reinforcement learning for planning based on the SMC perspective. We empirically demonstrated that our method achieves state of the art results on Mujoco. Our result suggest that planning can lead to faster learning in control tasks.

However, our particle-based inference method suffers some several shortcomings. First, we need many particles to build a good approximation of the posterior, and this can be computationally expensive since it requires to perform a forward pass of the policy, the value function and the model for every particle. Second, resampling can also have adverse effects, for instance all the particles could be resampled on the most likely particle, leading to a particle degeneracy. More advanced SMC methods dealing with this issue such as backward simulation (Lindsten et al., 2013) or Particle Gibbs with Ancestor Sampling (PGAS) (Lindsten et al., 2014) have been proposed and using them would certainly improve our results.

Another issue we did not tackle in our work is the use of models of the environment learned from data. Imperfect model are known to result in compounding errors for prediction over long sequences. We chose to re-plan at each time step (Model Predictive Control) as it is often done in control to be more robust to model errors. More powerful models or uncertainty modeling techniques can also be used to improve the accuracy of our planning algorithm. While the inference and modeling techniques used here could be improved in multiple ways, SMCP achieved impressive learning speed on complex control tasks. The planning as inference framework proposed in this work is general and could serve as a stepping stone for further work combining probabilistic inference and deep reinforcement learning.

### ACKNOWLEDGMENTS

The authors would like to thank Julien Cornebise, Philippe Beaudoin and Alexandre Bouchard-Côté for their interest and useful discussions. The authors would like to thank Stephanie Long, Anqi Xu, Kris Sankaran, Adrien Ali-Taïga, Rémi Le Priol, and Christian Rupprecht for reviewing an earlier version of the paper. The authors want to thank the Open Philanthropy Project, NSERC, Canada Research Chairs, CIFAR AI Chairs, National Science Foundation awards EHR-1631428 and SES-1461535 as well as NVIDIA for equipment.

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

# A APPENDIX

## A.1 ABBREVIATION AND NOTATION

Table A.1: Abbreviation

| | |
|---|---|
| SMCP: | Sequential Monte Carlo Planning. |
| SAC: | Soft Actor Critic. |
| CEM: | Cross Entropy Method. |
| RS: | Random Shooting. |
| MCTS: | Monte Carlo Tree Search. |
| SMC: | Sequential Monte Carlo. |
| SIR: | Sequential Importance Resampling. |
| SIS: | Sequential Importance Sampling. |
| IS: | Importance Sampling. |
| MPC: | Model Predictive Control |

Table A.2: Notation

| | | |
|---|---|---|
| $\mathbf{x}_{1:T}$ | $\triangleq$ | $\{\mathbf{s}_i, \mathbf{a}_i\}_{i=1}^{T}$ the state-action pairs. |
| $V$ | $\triangleq$ | value function. |
| $\mathcal{O}_t$ | $\triangleq$ | Optimality variable. |
| $p(\mathcal{O}_t|\mathbf{s}_t, \mathbf{a}_t)$ | $\triangleq$ | $\exp(r(\mathbf{s}_t, \mathbf{a}_t))$ Probability of a pair state action of being optimal. |
| $p_{\text{env}}$ | $\triangleq$ | Transition probability of the environment. Takes state and action $(\mathbf{s}_t, \mathbf{a}_t)$ as argument and return next state and reward $(\mathbf{s}_{t+1}, r_t)$. |
| $p_{\text{model}}$ | $\triangleq$ | Model of the environment. Takes state and action $(\mathbf{s}_t, \mathbf{a}_t)$ as argument and return next state and reward $(\mathbf{s}_{t+1}, r_t)$. |
| $w_t$ | $\triangleq$ | Importance sampling weight. |
| $p(\mathbf{x})$ | $\triangleq$ | Density of interest. |
| $q(\mathbf{x})$ | $\triangleq$ | Approximation of the density of interest. |
| $t \in \{1, \dots T\}$ | $\triangleq$ | time steps. |
| $n \in \{1, \dots N\}$ | $\triangleq$ | particle number. |
| $h$ | $\triangleq$ | horizon length. |

## A.2 THE ACTION PRIOR

The true joint distribution 2.1 in section 2.1 should actually be written:

$$
\begin{aligned}
p(\mathbf{x}_{1:T}, \mathcal{O}_{1:T}) &= \mu(\mathbf{s}_1) \prod_{t=1}^{T-1} p_{\text{env}}(\mathbf{s}_{t+1}|\mathbf{a}_t, \mathbf{s}_t) \prod_{t=1}^{T} p(\mathbf{a}_t) \exp\left(\sum_{t=1}^{T} r(\mathbf{s}_t, \mathbf{a}_t)\right) \\
&= \mu(\mathbf{s}_1) \prod_{t=1}^{T-1} p_{\text{env}}(\mathbf{s}_{t+1}|\mathbf{a}_t, \mathbf{s}_t) \exp\left(\sum_{t=1}^{T} r(\mathbf{s}_t, \mathbf{a}_t) + \log p(\mathbf{a}_t)\right)
\end{aligned}
$$

In Mujoco environments, the reward is typically written as

$$
r(\mathbf{s}_t, \mathbf{a}_t) = f(\mathbf{s}_t) - \alpha ||\mathbf{a}_t||_2^2
$$

where $f$ is a function of the state (velocity for HalfCheetah on Mujoco for example). The part $\alpha||\mathbf{a}_t||_2^2$ can be seen as the contribution from the action prior (here a gaussian prior). One can also consider the prior to be constant (and potentially improper) so that is does not change the posterior $p(\mathbf{x}_{1:T}|\mathcal{O}_{1:T})$.

## A.3  VALUE FUNCTION: BACKWARD MESSAGE

$$
\begin{aligned}
p(\mathcal{O}_{t+1:T}|\mathbf{x}_t) &= \int_{\mathbf{x}_{t+1}} p(\mathcal{O}_{t+1:T}, \mathbf{x}_{t+1}|\mathbf{x}_t)d\mathbf{x}_{t+1} \\
&= \int_{\mathbf{x}_{t+1}} p(\mathbf{x}_{t+1}|\mathbf{x}_t, \mathcal{O}_{t+1:T})p(\mathcal{O}_{t+1:T}|\mathbf{x}_{t+1})d\mathbf{x}_{t+1} \\
&= \int_{\mathbf{s}_{t+1}} p_{\text{env}}(\mathbf{s}_{t+1}|\mathbf{s}_t, \mathbf{a}_t) \left[ \int_{\mathbf{a}_{t+1}} p(\mathbf{a}_{t+1}|\mathbf{s}_{t+1}, \mathcal{O}_{t+1:T}) \exp Q(\mathbf{s}_{t+1}, \mathbf{a}_{t+1}) d\mathbf{a}_{t+1} \right] d\mathbf{s}_{t+1} \\
&= \int_{\mathbf{s}_{t+1}} p_{\text{env}}(\mathbf{s}_{t+1}|\mathbf{s}_t, \mathbf{a}_t) \exp\left(V(\mathbf{s}_{t+1})\right)d\mathbf{s}_{t+1} \\
&= \mathbb{E}_{\mathbf{s}_{t+1}|\mathbf{s}_t, \mathbf{a}_t}\left[\exp\left(V(\mathbf{s}_{t+1})\right)\right]
\end{aligned}
\tag{A.1}
$$

By definition of the optimal value function in (Levine, 2018).

## A.4  RECURSIVE WEIGHTS UPDATE

$$
\begin{aligned}
w_t &= \frac{p(\mathbf{x}_{1:t}|\mathcal{O}_{1:T})}{q(\mathbf{x}_{1:t})} \\
&= \frac{p(\mathbf{x}_{1:t-1}|\mathcal{O}_{1:T})}{q(\mathbf{x}_{1:t-1})} \frac{p(\mathbf{x}_t|\mathbf{x}_{1:t-1}, \mathcal{O}_{1:T})}{q(\mathbf{x}_t|\mathbf{x}_{1:t-1})} \\
&= w_{t-1} \cdot \frac{p(\mathbf{x}_t|\mathbf{x}_{1:t-1}, \mathcal{O}_{1:T})}{q(\mathbf{x}_t|\mathbf{x}_{1:t-1})} \\
&= w_{t-1} \frac{1}{q(\mathbf{x}_t|\mathbf{x}_{1:t-1})} \frac{p(\mathbf{x}_{1:t}|\mathcal{O}_{1:T})}{p(\mathbf{x}_{1:t-1}|\mathcal{O}_{1:T})}
\end{aligned}
$$

We use there the forward-backward equation 3.1 for the numerator and the denominator

$$
\begin{aligned}
&\propto w_{t-1} \frac{1}{q(\mathbf{x}_t|\mathbf{x}_{1:t-1})} \frac{p(\mathbf{x}_{1:t}|\mathcal{O}_{1:t})}{p(\mathbf{x}_{1:t-1}|\mathcal{O}_{1:t-1})} \frac{p(\mathcal{O}_{t+1:T}|\mathbf{x}_t)}{p(\mathcal{O}_{t:T}|\mathbf{x}_{t-1})} \\
&= w_{t-1} \frac{p(\mathbf{x}_t|\mathbf{x}_{1:t-1})}{q(\mathbf{x}_t|\mathbf{x}_{1:t-1})} p(\mathcal{O}_t|\mathbf{x}_t) \frac{p(\mathcal{O}_{t+1:T}|\mathbf{x}_t)}{p(\mathcal{O}_{t:T}|\mathbf{x}_{t-1})} \\
&= w_{t-1} \frac{p_{\text{env}}(\mathbf{s}_t|\mathbf{s}_{t-1}, \mathbf{a}_{t-1})}{p_{\text{model}}(\mathbf{s}_t|\mathbf{s}_{t-1}, \mathbf{a}_{t-1})} \frac{\exp(r_t)}{\pi_\theta(\mathbf{a}_t|\mathbf{s}_t)} \frac{\mathbb{E}_{\mathbf{s}_{t+1}|\mathbf{s}_t, \mathbf{a}_t}\left[\exp\left(V(\mathbf{s}_{t+1})\right)\right]}{\mathbb{E}_{\mathbf{s}_t|\mathbf{s}_{t-1}, \mathbf{a}_{t-1}}\left[\exp\left(V(\mathbf{s}_t)\right)\right]} \\
&= w_{t-1} \frac{p_{\text{env}}(\mathbf{s}_t|\mathbf{s}_{t-1}, \mathbf{a}_{t-1})}{p_{\text{model}}(\mathbf{s}_t|\mathbf{s}_{t-1}, \mathbf{a}_{t-1})} \mathbb{E}_{\mathbf{s}_{t+1}|\mathbf{s}_t, \mathbf{a}_t}\left[\exp\left(r_t - \log \pi_\theta(\mathbf{a}_t|\mathbf{s}_t) + V(\mathbf{s}_{t+1}) - \log \mathbb{E}_{\mathbf{s}_t|\mathbf{s}_{t-1}, \mathbf{a}_{t-1}}\left[\exp\left(V(\mathbf{s}_t)\right)\right]\right)\right]
\end{aligned}
\tag{A.2}
$$

## A.5  EXPERIMENT DETAILS

**Random samples:**  1000 transitions are initially collected by a random policy to pretrain the model and the proposal distribution. After which the agents start following their respective policy.

**Data preprocessing:**  We normalize the observations to have zero mean and standard deviation 1.

**Model Predictive Control:**  The model is used to predict the planning distribution for the horizon $h$ of $N$ particles. We then sample a trajectory according to its weight and return the first action of this trajectory. In our experiments, we fix the maximum number of particles for every method to 2500. For SMCP, the temperature and horizon length are described in Table A.3.

**Soft Actor Critic:** We used a custom implementation with a Gaussian policy for both the SAC baseline and the proposal distribution used for both versions of SMCP. We used Adam (Kingma & Ba, 2014) with a learning rate of 0.001. The reward scaling suggested by Haarnoja et al. (2018) for all experiments and used an implementation inspired by Pong (2018). We used a two hidden layers with 256 hidden units for the three networks: the value function, the policy and the soft Q functions.

**Model:** We train the model $p_{\text{model}}$ to minimize the negative log likelihood of $p(s_{t+1}|s_t + \Delta_t(s_t, a_t), \sigma_t(s_t, a_t))$. The exact architectures are detailed in Table A.3. We train the model to predict the distribution of the change in states and learn a deterministic reward function from the current state and predict the change in state. Additionally, we manually add a penalty on the action magnitude in the reward function to simplify the learning. At the end of each episode we train the model for 10 epochs. Since the training is fairly short, we stored every transitions into the buffer. The model is defined as:

$$\Delta s_t \sim p(\cdot|s_t, a_t) \tag{A.3}$$

$$r_t = g(s_t, \Delta s_t) - \alpha\|a\|^2 \tag{A.4}$$

where $\alpha$ was taken from the Mujoco gym environments. We used Adam (Kingma & Ba, 2014) with a learning rate of 0.001 and leaky ReLU activation function.

| Environment | Temperature | Horizon length | Number of Dense Layers | Layer Dimension |
|---|---|---|---|---|
| Hopper-v2 | 1 | 10 | 3 | 256 |
| Walker2d-v2 | 10 | 20 | 3 | 256 |
| HalfCheetah-v2 | 10 | 20 | 3 | 256 |

Table A.3: Hyperparameters for the experiments.

A.6 SEQUENTIAL IMPORTANCE SAMPLING PLANNING

---
**Algorithm 2** SMC Planning using SIS
---
1: **for** $t$ in $\{1, \ldots, T\}$ **do**
2:     $\{\mathbf{s}_t^{(n)} = \mathbf{s}_t\}_{n=1}^N$
3:     $\{w_t^{(n)} = 1\}_{n=1}^N$
4:     **for** $i$ in $\{t, \ldots, t + h\}$ **do**
5:         // Update
6:         $\{\mathbf{a}_i^{(n)} \sim \pi(\mathbf{a}_i^{(n)}|\mathbf{s}_i^{(n)})\}_{n=1}^N$
7:         $\{\mathbf{s}_{i+1}^{(n)}, r_i^{(n)} \sim p_{\text{model}}(\cdot|\mathbf{s}_i^{(n)}, \mathbf{a}_i^{(n)})\}_{n=1}^N$
8:         $\{w_i^{(n)} \propto w_{i-1}^{(n)} \cdot \exp\left(A(\mathbf{s}_i^{(n)}, \mathbf{a}_i^{(n)}, \mathbf{s}_{i+1}^{(n)})\right)\}_{n=1}^N$
9:     **end for**
10:     Sample $n \sim \text{Categorical}(w_{t+h}^{(1)}, \ldots, w_{t+h}^{(N)})$.
11:     // Model Predictive Control
12:     Select $\mathbf{a}_t$, first action of $\mathbf{x}_{t:t+h}^{(n)}$
13:     $\mathbf{s}_{t+1}, r_t \sim p_{\text{env}}(\cdot|\mathbf{s}_t, \mathbf{a}_t)$
14:     Add $(\mathbf{s}_t, \mathbf{a}_t, r_t, \mathbf{s}_{t+1})$ to buffer $\mathcal{B}$
15:     Update $\pi$, $V$ and $p_{\text{model}}$ with $\mathcal{B}$
16: **end for**
---

## A.7 Significance of the results

The significance of our results is done following guidelines from Colas et al. (2018). We test the hypothesis that the mean return of our method is superior to the one of SAC. We use 20 random seeds (from 0 to 19pro) for each method on each environment.

For this we look at the average return from steps 150k to 250k for SIR-SAC and SAC, and conduct a Welch's t-test with unknown variance. We report the $p$-value for each environment tested on Mujoco. A $p_{\text{val}} < 0.05$ usually indicates that there is strong evidence to suggest that our method outperforms SAC.

- `HalfCheetah-v2`: $p_{\text{val}} = 0.003$. There is very compelling evidence suggesting we outperform SAC.
- `Hopper-v2`: $p_{\text{val}} = 0.09$. There is no significant evidence suggesting we outperform SAC.
- `Walker2d-v2`: $p_{\text{val}} = 0.03$. There is compelling evidence suggesting we outperform SAC.

## A.8 Additional experimental results

### A.8.1 Effective Sample Size

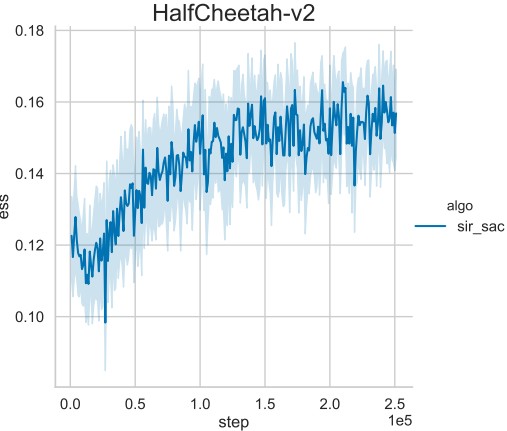

Figure A.1: Effective sample size for HalfCheetah. The shaded area represents the standard deviation over 20 seeds.

The values reported on Figure A.1 are the harmonic mean of the ratio of the effective sample size by the actual number of particles.

More precisely the values are

$$y_t = \Big( \prod_{i=1}^{h} \text{ESS}_i(t)/N \Big)^{1/h}$$

where $i$ is the depth of the planning, $N$ is the number of particles and

$$\text{ESS}_i(t) = \frac{(\sum_{n=1}^{N} w_{t+i}^{(n)})^2}{\sum_{n=1}^{N} (w_{t+i}^{(n)})^2}$$

We can see that as the proposal distribution improves the ESS also increases. The ESS on HalfCheetah is representative of the one obtained on the other environments. While these values are not high, we are still around $15\%$ thus we do not suffer heavily from weight degeneracy.

### A.8.2 Model loss

We also report the negative log likelihood loss of the environment's model during the training on Figure A.2.

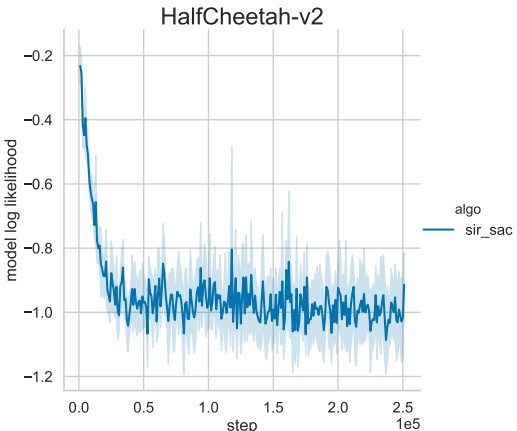

Figure A.2: Negative log likelihood for the model on HalfCheetah. The shaded area represents the standard deviation over 20 seeds.

