# OpenReview forum: "Probabilistic Planning with Sequential Monte Carlo methods"
_ICLR.cc/2019/Conference_

### Official Review · AnonReviewer1 · 2018-10-31
**Interesting preliminary work, but requires major revisions**

**Rating:** 5
**Confidence:** 4

**Review:**

The authors formulate planning as sampling from an intractable distribution motivated by control-as-inference, propose to approximately sample from the distribution using a learned model of the environment and SMC, then evaluate their approach on 3 Mujoco tasks. They claim that their method compares favorably to model-free SAC and to CEM and random shooting (RS) planning with model-based RL.

This is an interesting idea and an important problem, but there appear to be several inconsistencies in the proposed algorithm and the experimental results do not provide compelling support for the algorithm. In particular,

Levine 2018 explains that with stochastic transitions, computing the posterior leads to overly optimistic behavior because the transition dynamics are not enforced, whereas the variational bound explicitly enforces that. Is that an issue here?

The value function estimated in SAC is V^\pi the value function of the current policy. The value function needed in Sec 3.2 is a different value function. Can the authors clarify on this discrepancy?

The SMC procedure in Alg 1 appears to be incorrect. It multiplies the weights by exp(V_{t+1}) before resampling. This needs to be accounted for by setting the weights to exp(-V_{t+1}) instead of uniform. See for example auxiliary particle filters.

The experimental section could be significantly improved by addressing the following points:
* How was the planning horizon h chosen? Is the method sensitive to this choice? What is the model accuracy?
* Does CEM use a value function? If not, it seems like a reasonable baseline to consider CEM w/ a value function to summarize the values beyond the planning horizon. This will evaluate whether SMC or including the value function is important.
* Comparing to state-of-the-art model-based RL (e.g., one of Chua et al. 2018, Kurutach et al. 2018, Buckman et al. 2018).
* How were the task # of steps chosen? They seem arbitrary. What is the performance at 1million and 5million steps?
* Was SAC retuned for this small number of samples/steps?
* Clarify where the error bars come from in Fig 5.2 in the caption.
At the moment, SMCP is within the error bars of a baseline method.

Comments:

In the abstract, the authors claim that the major challenges in planning are: 1) model compounding errors in roll-outs and 2) the exponential search space. Their method only attempts to address 2), is that correct? If so, can the authors state that explicitly.

Recent papers (Chua et al. 2018, Kurutach et al. 2018, Buckman et al. 2018, Ha and Schmidhuber 2018) all show promising model-based results on continuous state/action tasks. These should be mentioned in the intro.

The connection between Gu et al.'s work on SMC and SAC was unclear in the intro, can the authors clarify?

For consistency, ensure that sums go to T instead of \infty.

I found the discussion of SAC at the end of Sec 2.1 confusing. As I understand SAC, it does try to approximate the gradient of the variational bound directly. Can the authors clarify what they mean?

At the end of Sec 2.2, the authors claim that the tackle the particle degeneracy issue (a potentially serious issue) by "selecting the temperature of the resampling distribution to not be too low." I could not find further discussion of this anywhere in the paper or appendix.

Sec 3.2, mentions an action prior for the first time. Where does this come from?

Sec 3.3 derives updates assuming a perfect model, but we learn a model. What are the implications of this?

Please ensure the line #'s and the algorithm line #'s match.

Model learning is not described in the main text though it is a key component of the algorithm. The appendix lacks details (e.g., what is the distribution used to model the next state?) and contradicts itself (e.g., one place says 3 layers and another says 2 layers).

In Sec 4.1, a major difference between MCTS and SMC is that MCTS runs serially, whereas SMC runs in parallel. This should be noted and then it's unclear whether SMC-Planning should really be thought of as the maximum entropy tree search equivalent of MCTS.

In Sec 4.1, the authors claim that Alpha-Go and SMCP learn proposals in similar ways. However, SMCP minimizes the KL in the reverse direction (from stated in the text). This is an important distinction.

In Sec 4.3, the authors note that Gu et al. learn the proposal with the reverse KL from SMCP. VSMC (Le et al. 2018, Naesseth et al. 2017, Maddison et al. 2017) is the analogous work to Gu et al. that learn the proposal using the same KL direction as SMCP. The authors should consider citing this work as it directly relates to their algorithm.

In Sec 4.3, the authors claim that their direction of minimizing KL is more appropriate for exploration. Gu et al. suggest the opposite in their work. Can the author's justify their claim?

In Sec 5.1, the authors provide an example of SMCP learning a multimodal policy. This is interesting, but can the authors explain when this will be helpful?

====

11/26
At this time, the authors have not responded to reviews. I have read the other reviews. Given the outstanding issues, I do not recommend acceptance.

12/7
After reading the author's response, I have increased my score. However, baselines that establish the claim that SMC improves planning which leads to improved control are missing (such as CEM + value function). Also, targeting the posterior introduces an optimism bias that is not dealt with or discussed.

---

> ### Author Response · Authors · 2018-11-27
> **Answer to reviewer 1/3**
>
> We would like to thank the reviewer for this very thorough review. We believe that these comments are making the paper clearer and stronger.
>
> 1) “[...] the experimental results do not provide compelling support for the algorithm.“
>
> We agree the initial results were not compelling in that regard. We have updated the results and we now believe the performance of our planning method appears clearly. We used 20 seeds and also added a significance test following guidelines by Colas et al 2018 in Appendix A.7. We furthermore added more experimental details in the Appendix A.5 and A.8
>
> 2) “Levine 2018 explains that with stochastic transitions, computing the posterior leads to overly optimistic behavior because the transition dynamics are not enforced, whereas the variational bound explicitly enforces that. Is that an issue here?”
>
> Our model is trained by maximum likelihood as in Chua et al. 2018 only from data, separately from the policy and planning. Thus, the policy has no control over the system dynamics, hence the model is not encouraged to yield over-optimistic transitions.  We have added details about the model training procedure in the experiments section and have update our pseudo-code for clarity.
>
> 3) “The value function estimated in SAC is V^\pi the value function of the current policy. The value function needed in Sec 3.2 is a different value function. Can the authors clarify on this discrepancy?”
>
> Indeed. However as we do not have access to the optimal value function, we use the current value function of SAC as a proxy. As the SAC-policy will converge to a policy closer to optimality, so will its value function. Therefore we think this is a sensible practical choice, and this is similar to what is done in actor-critic methods for instance.
>
> 4) “The SMC procedure in Alg. 1 appears to be incorrect. It multiplies the weights by exp(V_{t+1}) before resampling. This needs to be accounted for by setting the weights to exp(-V_{t+1}) instead of uniform. See for example auxiliary particle filters.”
>
> Yes, there was indeed an issue with the weight update that we have now fixed and it does indeed align with your intuition.
> To be precise, we believe the weight update should be done pour multiplying with the previous weight by exp(r - log pi + V’ -log E_{s_t | a_t-1 s_t-1} exp V(s_t)).
> We thought (wrongly) that the log-expectation-exp was equal to the normalization constant when normalizing the weights, thus redundant. However this normalization constant takes its expectation under the states the particles are in at time t rather than the transition dynamics as it should be done.
> By fixing the update, we now believe we have the right formula, and this allows us to have an unweighted empirical estimate of the posterior.
> This is indeed similar in spirit to the auxiliary particle filter, we thank you for the reference and for pointing out the issue, it helped us derive the right update formula.
>
> 5) “How was the planning horizon h chosen? Is the method sensitive to this choice? What is the model accuracy?” + “At the end of Sec 2.2, the authors claim that the tackle the particle degeneracy issue (a potentially serious issue) by "selecting the temperature of the resampling distribution to not be too low." I could not find further discussion of this anywhere in the paper or appendix.”
>
> We did not do any extensive hyperparameter search in the beginning. We tried mostly temperatures in the range [1-10]. We checked the ESS while training to make sure we did not have any weight degeneracy issue. See A.8 for a plot of the ESS during training.
> We have tried horizons from 5 to 50, and while the performance is pretty stable across this range of horizons, h~20 seems a good value to work with for Walker2d and HalfCheetah. Hopper was more challenging and we found out that typically shorter horizons worked marginally better.
> The path degeneracy is indeed a very serious issue, and we definitely suffer from it even when tuning the temperature. While some modern smoothing algorithms like Particle Gibbs with Ancestor Sampling can alleviate it, our goal in this work is to introduce a new simple and motivated way of doing planning rather than obtaining the best performance possible.

---

> > ### Comment · AnonReviewer1 · 2018-12-07
> > **RE:**
> >
> > The optimism bias stems from targeting the posterior, and is not due to errors in modeling the transitions.

---

> > > ### Author Response · Authors · 2018-12-07
> > > **Clarifications**
> > >
> > > 1)  “However, baselines that establish the claim that SMC improves planning which leads to improved control are missing (such as CEM + value function). “
> > >
> > > We understand that CEM with a value function would be an interesting baseline, but we are not aware of any work that introduced what you are mentioning. Could you point us toward relevant work using CEM with a value function? For example, even the most recent work we could find using CEM (Hafner 2018) does not use a value function.
> > >
> > > For instance, it is unclear to us how the value function should be learned. The most natural way to learn a value function, would be to do it online (ie learn the value function induced by the non-parametric CEM policy). Another alternative would be to learn a value function offline, but this would would be expensive since it would require to do a full planning step i.e. querying the generative model for multiple steps and then correcting the action chosen. Then we could either correct the expectation with importance sampling or use a Q-function similar to SAC.
> > >
> > > We think there are many ways this could be designed, leading to various performances and behaviors: this is a very interesting direction, but we believe this would require a full paper rather than being introduced as a baseline.
> > >
> > > In any case, we believe we have indeed very strong evidence to support our claims that SMC improves the sample efficiency of the model free proposal (section 5.2 experiments were done with 20 seeds following best practices from Henderson 2017 and Colas 2018, which is greatly superior to what is usually done in the field).
> > >
> > > 2) “The optimism bias stems from targeting the posterior, and is not due to errors in modeling the transitions”
> > >
> > > Are your referring to “exact inference in the graphical model produces an “optimistic” policy that assumes some degree of control over the system dynamics.” - Levine 2018, section 5.4?
> > > In our case, the model is NOT trained jointly with the policy (only from buffer data), so the policy does not assume any control on the system’s dynamics, thus our posterior is not overly optimistic.

---

> > > > ### Comment · AnonReviewer1 · 2018-12-11
> > > > **RE:**
> > > >
> > > > 1. As done in your proposal, can the value function from SAC be used?
> > > >
> > > > 2. The optimism bias does not stem from model error. The exact posterior with a perfect model suffers  from optimism bias in stochastic environments. This is what is meant by Levine 2018.

---

> > > > > ### Author Response · Authors · 2018-12-12
> > > > > **About the optimism bias**
> > > > >
> > > > > Optimism bias
> > > > > ===
> > > > >
> > > > > First, thank you for the fast answer, this is really appreciated. We furthermore had the opportunity recently to discuss with various researchers about our work and this question was raised several times. Therefore we are now convinced that it should be discussed on the final version of paper no matter what conclusion our discussion reaches.
> > > > >
> > > > > For clarity p(s'|s, a) is what we called p_env(s'|s, a) in our work while q(s'|s, a) would be p_model(s'|s, a).
> > > > > (we were actually wondering if that would be clearer to use in the paper as well.)
> > > > >
> > > > > How we understand the optimism bias
> > > > > ---
> > > > > When optimizing the the posterior with a variational proposal q e.g KL(q(x)||p(x|O)), we obtain the objective described in Levine 2018, sec 2.4, eq 10. It contains the expectation under q of the reward, the expectations under q of log p(s'|s, a) and the entropy of q.
> > > > > The important point is that we have some divergences of the transitions given by q and by p.
> > > > >
> > > > > However, if we maximize q with this objective, then we will learn a wrong transition model that assumes overly optimistic transitions.
> > > > > Indeed, this is because the reward signal/optimality has been used implicitly to train the transition model q(s'|s,a). The transition model is actually trying to match p(s' | s, a, O) instead of p(s' | s, a) [Levine 2018, sec 2.4, eq 9]. This is due to the fact that the factorizations of p and q are different. This learned transition model is wrong and we believe this is what is called the optimism bias.
> > > > >
> > > > > This is corroborated by [Levine 2018 sec 3]:
> > > > > > The problematic nature of the maximum entropy framework in the case of stochastic dynamics, discussed in Section 2.3 and Section 2.4, in essence amounts to an assumption that the agent is allowed to control both its actions and the dynamics of the system in order to produce optimal trajectories, but its authority over the dynamics is penalized based on deviation from the true dynamics.
> > > > >
> > > > >
> > > > > Why we believe we don't suffer from it and why we think Levine 2018 corroborates our view
> > > > > ---
> > > > >
> > > > > The solution proposed by Levine is to fix q(s'|s,a) to p(s'|s,a) [Levine 2018 sec 3.1], that way, q(s'|s,a) does not "see" the reward/optimality and thus can't be over-optimistic. This can be seen as a type of variational inference as well where some structure (here q(s'|s,a) == p(s'|s,a)) is forced into the variational distribution [Levine 2018 sec 3.2].
> > > > >
> > > > > More generally, the issue we have to avoid is that q(s'|s,a) should NOT be trained to match p(s'|s, a, O) as jointly optimizing everything would do.
> > > > > In our case, we specifically force q(s'|s, a) to match p(s'|s, a) by doing it explicitly and training q(s'|s, a) to match p(s'| s, a) by MLE.
> > > > >
> > > > > Filtering, control and the posterior
> > > > > ---
> > > > > More generally, targeting a posterior like we do seems to be very widespread and established in the filtering and control communities. For instance a Kalman smoother estimates perfectly the posterior p(x_{1:T} | y_{1:T}) for linear-gaussian systems.
> > > > > Do you believe that these methods also suffer from the optimism bias?
> > > > > Our understanding is that they don't as the transition model (even when imperfect) is not trained to optimize the posterior but either known, modeled by hand or estimated by MLE from transition data (as we do).
> > > > >
> > > > >
> > > > > I'd like to re-emphasize that we are open to the discussion.
> > > > > - If you can convince us that we do suffer from the optimism bias, we'll gladly add a subsection discussing it and why we think our method still works or how we could improve on it maybe.
> > > > > - If we can convince you that this is not an issue in this work, we believe we should still state in our paper why this is the case.
> > > > >
> > > > > Please feel free to detail your thoughts and tell us exactly where you disagree with us.

---

> > > > > > ### Comment · AnonReviewer1 · 2018-12-12
> > > > > > **RE:**
> > > > > >
> > > > > > Yes, your understanding of the optimism bias is incorrect.
> > > > > >
> > > > > > The problem does not stem from inaccuracies in q, in fact, when q = p_env, the optimism bias is present. As you defined in your paper, p(x | O) \propto p_env(x) \exp(\sum r_t). The problem is that p(x | O) incorporates exp(\sum r_t) which biases samples of x towards states and actions with high reward. This is fine for actions, but causes an optimism bias for state transitions. Note that p(x | O) uses the real environment model and does not even depend on q, yet there is still optimism bias. As a result, under the posterior p(x | O), p(s' | s, a, O) != p_env(s' | s, a) which is the optimism bias meant by Levine.
> > > > > >
> > > > > > For LQR systems, we compute p(x | y) w/o the reward (ie., no exp(\sum r_t)) term. As a result, there is no optimism bias.

---

> > > > > > > ### Author Response · Authors · 2018-12-18
> > > > > > > **Answer**
> > > > > > >
> > > > > > > Thank you for the more detailed answer, we think we finally understood the source of our disagreement. We believe we do not have the same definition of optimism bias, and while we do not suffer from any agent's delusion about the world, we do suffer from an overestimation bias of the mean return.
> > > > > > >
> > > > > > > 1. In brief, the issue we believe you are talking about is the objective itself and thus intrinsic to the posterior.
> > > > > > > Indeed, we do maximize log-Expectation-exp(Return) which is an upper bound on the expected return. Thus maximizing our objective might not mean that we have a good expected return. This is common in many control as inference methods.
> > > > > > > We are not certain what terminology is used in RL, but we would rather call that an overestimation bias.
> > > > > > >
> > > > > > > 2. The optimism bias, even in psychology, is a delusion of the agent about the world, ie the agent believes the world will lead to unrealistically more desirable outcomes ( ie q(s'|s, a) = p(s'|s,a,O) instead of p(s'|s,a) ).
> > > > > > > This is actually the issue we were mentioning from the beginning and we explained why we do not suffer from it.
> > > > > > >
> > > > > > > While point 2 is not an issue for us, point 1 as you raised is indeed one. We will add a paragraph in the final version of the paper to explain the distinction.
> > > > > > > Why don't we suffer heavily from 1 then?
> > > > > > > Our guess is that Mujoco is very close to deterministic and our model of the world learns very rapidly to predict the next state with a very low variance, thus we believe our transitions are close to deterministic, making this less of an issue.

---

> > > > > > > > ### Comment · AnonReviewer1 · 2019-01-02
> > > > > > > > **RE: Answer**
> > > > > > > >
> > > > > > > > Great.
> > > > > > > >
> > > > > > > > 1. Yes, this is the optimism issue raised in Levine 2018. Most control as inference methods use a structured variational objective, which fundamentally gets at this optimism bias.
> > > > > > > >
> > > > > > > > 2. I'm not familiar with RL algorithms that suffer from this issue. If you have references, it would be great to add them to the discussion of related works.

---

> > > > > ### Author Response · Authors · 2018-12-12
> > > > > **About CEM+value**
> > > > >
> > > > > CEM+value baseline
> > > > > ===
> > > > >
> > > > > Yes this is an option!
> > > > > So the proposed algorithm would be to use regular CEM, but for the optimization to maximize \sum_i=1^h r_i + V_{t+h+1} where V is the value function from SAC instead of just \sum_i=1^h r_i.
> > > > >
> > > > > We are currently running it, and we do have some preliminary results. It does seem to do better on Hopper (some seeds seem to be around 1000 of return which vanilla CEM never did, while others are still very low), we don't see improvements over CEM for HalfCheetah and Walker2d for now.
> > > > >
> > > > > However just a few points:
> > > > > - It seems to have some instabilities eg the value and Q-loss seem to always diverge (>>> billions while for (SIR)-SAC it is around 1 to 10). We believe it may be because it is using the value function from SAC while using the policy from CEM which could be really different. In our work, we could expect our planning policy to be closer to SAC policy as SAC policy was actually used as a proposal.
> > > > > - It is probably possible to augment CEM with a value function in a principled and more stable way, but we think it is a contribution in itself and should be explored in a full paper.

---

> > > > > > ### Author Response · Authors · 2018-12-18
> > > > > > **Final results on CEM+value function**
> > > > > >
> > > > > > Our final results with CEM+value function show no improved performance overall over vanilla CEM. This seems mainly due to the fact that the CEM policy and the SAC value function do not match and our value/Q losses diverge.

---

> ### Author Response · Authors · 2018-11-27
> **Answer 2/3**
>
>
> 6) “How were the task # of steps chosen? They seem arbitrary. What is the performance at 1million and 5million steps?”
>
> As stated in the conclusion, our algorithm is expensive. Given that we train the model, the SAC networks (policy, value and Q functions), and we perform a full planning a each time step (MPC), training for 250k steps already takes a few days.
> We decided to allocate our computing resources on producing more seeds rather than longer runs. It should be noted however that we do not expect our algorithm to keep outperforming SAC in the long run. We believe this is a behavior to be expected when planning with imperfect models, in the long run, the model-free method will find a good policy while the planning part will still suffer from model errors. We think this is also the case for humans; when confronted with a new situation we tend to plan, but as we become more familiar with it, our reflexes/habitus are more accurate.
> As a solution, we could also learn when and how long to plan, but we believe this is out of scope for this work.
>
>
> 7) “Was SAC retuned for this small number of samples/steps?”
>
> No, it was not, we took the default values from the SAC paper. However we think it is fair since we use the exact same version of SAC for our proposal distribution and thus the only difference is from the planning algorithm.
>
> 8) “Clarify where the error bars come from in Fig 5.2 in the caption.”
> Yes we have added clarification. The error bars are 1 standard deviation from the mean with 20 seeds for each algorithm. This is the default setting for the confidence interval computation with the seaborn package.
>
> 9) “In the abstract, the authors claim that the major challenges in planning are: 1) model compounding errors in roll-outs and 2) the exponential search space. Their method only attempts to address 2), is that correct? If so, can the authors state that explicitly.”
>
> You are correct, we reformulate the introduction to clearly state the problem we are tackling: search algorithm. We do acknowledge that this is a very important issue -but that is not part of our contribution- in the related work and conclusion sections.
>
> 10) “I found the discussion of SAC at the end of Sec 2.1 confusing. As I understand SAC, it does try to approximate the gradient of the variational bound directly. Can the authors clarify what they mean?”
>
> We clarified the discussion. We think the distinction is mostly that a policy gradient algorithm would use a Monte Carlo return while SAC uses soft value functions and the policy is taken to be the Boltzmann distribution over the soft-Q values. This discussion was inspired by Section 4.2 of Levine (2018).
>
> 11) “The connection between Gu et al.'s work on SMC and SAC was unclear in the intro, can the authors clarify?”
>
> We think this discussion is actually more adapted for the related work section. There, we have now clarified the connection between Gu’s work on SMC and SAC.
>
> 12)  “In Sec 4.1, a major difference between MCTS and SMC is that MCTS runs serially, whereas SMC runs in parallel. This should be noted and then it's unclear whether SMC-Planning should really be thought of as the maximum entropy tree search equivalent of MCTS.” + “In Sec 4.1, the authors claim that Alpha-Go and SMCP learn proposals in similar ways. However, SMCP minimizes the KL in the reverse direction (from stated in the text). This is an important distinction.” + “In Sec 4.3, the authors note that Gu et al. learn the proposal with the reverse KL from SMCP. VSMC (Le et al. 2018, Naesseth et al. 2017, Maddison et al. 2017) is the analogous work to Gu et al. that learn the proposal using the same KL direction as SMCP. The authors should consider citing this work as it directly relates to their algorithm.” + “In Sec 4.3, the authors claim that their direction of minimizing KL is more appropriate for exploration. Gu et al. suggest the opposite in their work
>
> The reviewer correctly pointed out some inconsistencies and vagueness in the related work section. We decided to rewrite it concisely and only focus on pointing toward relevant work to ours.

---

> ### Author Response · Authors · 2018-11-27
> **Answer 3/3**
>
>
> 13) “Sec 3.2, mentions an action prior for the first time. Where does this come from?”
>
> This action prior comes from the factorization of the HMM model in section 2.1 (it is typically considered constant or already included in the reward). We follow the notation of Levine 2018 that omits it for conciseness. We decided to add a footnote on eq 2.1 for clarity as well as a section in the Appendix A.2.
>
> 14) “Sec 3.3 derives updates assuming a perfect model, but we learn a model. What are the implications of this?”
>
> This is a necessary assumption that most planning algorithm (CEM, LQR…) make. Implications of this assumption are model compounding errors on the plan. To be more robust to model errors, it is typical to replan a each time step (Model Predictive Control) as we do. We added some clarification in this subsection.
>
> 15) “Please ensure the line #'s and the algorithm line #'s match.”
>
> We have updated the algorithm section. Now the lines should match and the algorithm in written in a more comprehensive way.
>
> 16) “Does CEM use a value function? If not, it seems like a reasonable baseline to consider CEM w/ a value function to summarize the values beyond the planning horizon. This will evaluate whether SMC or including the value function is important.“
>
> We think it is fair to compare to it as it is. Indeed, CEM is a method that has been used successfully in multiple settings e.g. Tetris and it is the default algorithm for planning in the deep RL community (e.g. Chua 2018) and is a baseline algorithm for us.
> Moreover, as we do not do any learning in the toy example, SMCP does not use a value function. Even then, we see that our algorithm can handle multimodality while CEM cannot.
>
> 17) “Model learning is not described in the main text though it is a key component of the algorithm. The appendix lacks details and contradicts itself.” + “Comparing to state-of-the-art model-based RL.”
>
> We corrected inconsistencies and added details. Note that we used a fairly standard probabilistic model (gaussian likelihood) and focus most of the space to describe our contribution: the planning algorithm, since any good model would work well.
> These work are indeed relevant, but also complementary to ours. For example, Model Ensemble could potentially improve our results and those of the planning baselines. We added references to these papers in the text.
>
> 18) “In Sec 5.1, the authors provide an example of SMCP learning a multimodal policy. This is interesting, but can the authors explain when this will be helpful?”
>
> RL algorithms are known to suffer from a mode seeking behavior and often only discover suboptimal solutions. We believe the ability to handle multimodality could help discovering new solutions to a task.

---

### Official Review · AnonReviewer3 · 2018-10-31
**More work/evaluation on the SMC part needed**

**Rating:** 6
**Confidence:** 4

**Review:**

This paper proposes a sequential Monte Carlo Planning algorithm that depicts planning as an inference problem solved by SMC. The problem is interesting and the paper has a nice description of the related work. In terms of the connection between the the problem and Bayesian filtering as well as smoothing, the paper has novelty there. But it is unclear to me how the algorithm proposed is applicable in complex continuous tasks as claimed.

In the introduction, the authors wrote that "We design a new algorithm, Sequential Monte Carlo Planning (SMCP), by leveraging modern methods in Sequential Monte Carlo (SMC), Bayesian smoothing, and control as inference". From my understanding, the SMC algorithm adopted is the bootstrap particle which is the simplest and earliest SMC algorithm adopted. The Bayesian smoothing algorithm described is also standard. I do not see the modern parts of these algorithms.

The experiment section reports the return, but it is unclear to me how the SMC algorithm in this case. For example, what is the effective sample size (ESS) in these settings?

The experiment described seems to be a 2-dimensional set up. How does the algorithm perform with a high-dimensional planning problem?

---

> ### Author Response · Authors · 2018-11-27
> **Answer to the reviewer**
>
> We would like to thank the reviewer for the correction and added additional experimental details to better understand the behaviour of the method.
>
> 1) “[...] the SMC algorithm adopted [...] is the simplest and earliest SMC algorithm adopted.[...]. I do not see the modern parts of these algorithms.”
>
> This is fair point, we corrected the sentence.
>
> 2) “The experiment section reports the return, but it is unclear to me how the SMC algorithm in this case. For example, what is the effective sample size (ESS) in these settings?”
>
> In this case, the SMC algorithm now clearly outperforms the SAC baseline as you can see in the updated version of the plot. Furthermore, we have added a new section in the Appendix A.8 describing the evolution of the ESS during training. While it is not very high, is is usually around 15% of the sample size, which we believe is reasonable so that we do not suffer heavily from weight degeneracy.
>
> 3) “But it is unclear to me how the algorithm proposed is applicable in complex continuous tasks as claimed.”
>  And “The experiment described seems to be a 2-dimensional set up. How does the algorithm perform with a high-dimensional planning problem?”
>
> Yes, the 2d experiment is merely illustrative, the complex continuous tasks mentioned are illustrated with the experiments on Mujoco in subsection 2 of the experiments. In section 5.2, we have updated our performance results on the 3 classic Mujoco environments. Their respective state/action dimensions are:
> Walker2d-v2, state (17,), action (6,)
> Hopper-v2, state (11,), action (3,)
> HalfCheetah-v2, state (17,), action (6,)
>
> Still, we have removed the mention of “high dimensional” as control tasks (ie Mujoco), while complex, are maybe not what the statistical community would call “high dimensional”. Also, vanilla particle filters are known to suffer from the curse of dimensionality, especially if the proposal is poor.
> A solution we leave to future work would be to do the planning in latent space, in that case our method could scale even with very high dimensional inputs.

---

> ### Author Response · Authors · 2018-12-12
> **Additional feedback**
>
> Hello,
>
> We believe that we have addressed the points raised in your review (notably ESS plots + complex experiments).
> Did you also have time to look at the updated version of the paper?
>
> Looking forward to hearing from you soon,
> Thank you.

---

### Official Review · AnonReviewer2 · 2018-11-01
**Sequential Monte Carlo (SMC) has since its inception some 25 years ago proved to be a powerful and generally applicable tool. The authors of this paper continue this development in a very interesting and natural way by showing how SMC can be used to solve challenging planning problems. This is a enabled by reformulating the planning problem as an inference problem via the recent trend referred to as "control as inference".**

**Rating:** 8
**Confidence:** 4

**Review:**

Sequential Monte Carlo (SMC) has since its inception some 25 years ago proved to be a powerful and generally applicable tool. The authors of this paper continue this development in a very interesting and natural way by showing how SMC can be used to solve challenging planning problems. This is a enabled by reformulating the planning problem as an inference problem via the recent trend referred to as "control as inference". While there is unfortunately no real world experiments, the simulations clearly illustrate the potential of the approach.
While the idea of viewing control as inference is far from new the idea of using SMC in this context is clearly novel as far as I can see. Well, there has been some work along the same general topic before, see e.g.
Andrieu, C., Doucet, A., Singh, S.S., and Tadic, V.B. (2004). Particle methods for change detection, system identification, and contol. Proceedings of the IEEE, 92(3), 423–438.
However, the particular construction proposed in this paper is refreshingly novel and interesting. Hence, I view the specific idea put fourth in this paper as highly novel. The general idea of viewing control as inference goes far back and there are very nice dual relationships between LQG and the Kalman filter established and exploited long time ago.

The authors interprets "control as inference" as viewing the planning problem as a simulation exercise where we aim to approximate the distribution of optimal future trajectories. A bit more specifically, the SMC-based planning proposed in the paper stochastically explores the most promising trajectories in the tree and randomly removes (via the resampling operation) the less promising branches. Importantly there are convergence guarantees via the use of SMC. The idea is significant in that it opens up for the use of the by now strong SMC body of methods and analysis when it comes to challenging and intractable planning problems. I foresee many interesting developments to follow in the direction layed out by this paper.

When it comes to your SMC algorithm you will suffer from path degeneracy (as all SMC algorithms does, see e.g. Figure 1 in https://arxiv.org/pdf/1307.3180.pdf) and if h is large I think this can be a problem for you. However, this can easily be fixed via backward simulation. For an overview of backward simulation see
Lindsten, F. and Schon, T. "Backward simulation methods for Monte Carlo statistical inference". Foundations and Trends in Machine Learning, 6(1):1-143, 2013.

I am positive to this paper (clearly reveled by my score as well), but there are of course a few issues as well:
1. There are no theoretical results on the properties of the proposed approach. However, given the large body of literature when it comes to the analysis of SMC methods I would expect that you can provide some results via the nice bridge that you have identified.
2. Would this be possible to implement in a real-world setting with real-time requirements?
3. A very detailed question when it comes to Figure 5.2 (right-most plot), why is the performance of your method significantly degraded towards the end? It does recover indeed, but I still find this huge dip quite surprising.

Minor details:
* The initial references when it comes to SMC are wrong. The first papers are:
N.J. Gordon, D. Salmond and A.F.M. Smith, Novel approach to nonlinear/non-Gaussian Bayesian state estimation, IEE Proc. F, 1993
L. Stewart, P. McCarty, The use of Bayesian Belief Networks to fuse continuous and discrete information for target recognition and discrete information for target recognition, tracking, and situation assessment, in Proc. SPIE Signal Processing, Sensor Fusion and Target Recognition,, vol. 1699, pp. 177-185, 1992.
 G. Kitagawa, Monte Carlo filter and smoother for non-Gaussian nonlinear state-space models, JCGS, 1996
* When it comes to the topic of learning a good proposal for SMC with the use of variational inference the authors provide a reference to Gu et al. (2015) which is indeed interesting and relevant in this respect. However, on this hot and interesting topic there has recently been several related papers published and I would like to mention:
C. A. Naesseth, S. W. Linderman, R. Ranganath, D. M. Blei, Variational Sequential Monte Carlo. Proceedings of the 21st International Conference on Artificial Intelligence and Statistics, Lanzarote, Spain, April 2018.
C. J. Maddison, D. Lawson, G. Tucker, N. Heess, M. Norouzi, A. Mnih, A. Doucet, and Y. Whye Teh. Filtering variational objectives. In Advances in Neural Information Processing Systems, 2017.
T. A. Le, M. Igl, T. Jin, T. Rainforth, and F. Wood. AutoEncoding Sequential Monte Carlo. arXiv:1705.10306, May 2017.

I would like to end by saying that I really like your idea and the way in which you have developed it. I have a feeling that this will inspire quite a lot of work in this direction.

---

> ### Author Response · Authors · 2018-11-27
> **Answer to the reviewer**
>
> We would like to thank the reviewer for the encouraging comments and important references.
>
> “When it comes to your SMC algorithm you will suffer from path degeneracy. [...] However, this can easily be fixed via backward simulation [...]”
>
> Yes, we thank you for the suggestion. Particle Gibbs with Ancestral Sampling had been also brought to our attention to tackle this issue, but we choose to keep it simple in this work to focus more on introducing the idea rather than on getting the best results.
>
> 1) “There are no theoretical results on the properties of the proposed approach. However, given the large body of literature when it comes to the analysis of SMC methods I would expect that you can provide some results via the nice bridge that you have identified.”
>
> We believe our method is grounded when p_model = p_env and we have access to the optimal value function.
> However in most RL settings, both these assumptions are violated and it lessens the impact of the analysis.
> A very interesting theoretical analysis we wish to make is to look if we can still provide some guarantees when the model and value function are approximately optimal, but a full theoretical study is still upcoming and out of scope of this paper.
>
> 2) “Would this be possible to implement in a real-world setting with real-time requirements?”
>
> We think it is possible if we replan every few steps instead of every step and keep a reasonable number of particles. Several methods bringing SMC methods to real-time systems exist. For instance, for embedded systems with real-time constraints, a FPGA implementation of SMC has been proposed (Ndeved et al, 2014, https://www.sciencedirect.com/science/article/pii/S1474667016429812).
> We also believe that additionally to a good search algorithm, we need to learn good representations (eg if the input is an image) and plan in the latent space.
>
> 3) “A very detailed question when it comes to Figure 5.2 (right-most plot), why is the performance of your method significantly degraded towards the end? It does recover indeed, but I still find this huge dip quite surprising.”
>
> Indeed. We had more time to investigate this during the review period and we realized that some of our jobs were killed around step 40k. We have since rerun all our experiments and closely monitored that no such thing happened again. We are now confident our updated results are much stronger and show with high confidence the real performance of our method.

---

> > ### Comment · AnonReviewer2 · 2018-12-08
> > **Thank you for these clarifications.**
> >
> >  Glad to hear that the experiments are now corrected. My rating remain the same as before.

---

### Author Response · Authors · 2018-11-27
**Overview of the changes**

We would like first to thank all reviewers for their work. We did a major revision of the paper based on the issues pointed out. We believe this current form is now much clearer and stronger and addresses the points raised by the reviewers.

Outline of the revisions:
- Simplified the abstract and clarified the introduction.
- Fixed small typos and inaccuracies in section 2 (Background).
- We reworked section 3.3 and 3.4 (SMCP) and fixed an issue in the weight update.
- We added new strong and significant experimental results on Mujoco.
- We reworked and wrote a more comprehensive section 5 (Related work) and discussed relevant papers, such as the ones pointed out by the reviewers.
- Appendices: Included additional details and experimental figures.

---

### Public Comment · (anonymous) · 2018-12-17
**Some questions on your work.**

Hi, I think your work is interesting and have some questions as a reader of your work.

1. I cannot figure out how the i.i.d. prior for the action sequences, i.e., \prod_{t=1}^T p(a_t), can be used. I also checked Sergey Levine's tutorial and review on "RL as Inference", but i.i.d. action sequences are not shown in that tutorial. Would you please clarify this part? Personally, I think this part is quite weird.

2. Any plan to open your source code?

3. I wonder whether you've done a wall-clock-time comparison between model-free RL, e.g., SAC, and your work.

---

### Public Comment · ~Nando_de_Freitas1 · 2019-02-27
**Related references on sequential Monte Carlo and other inference methods for planning**

The following are related references on Sequential Monte Carlo methods for planning, and other Monte Carlo and EM methods for planning, which the authors and readers might find useful to know about:

Hoffman, M. W., Doucet, A., de Freitas, N., & Jasra, A. (2007). On solving general state-space sequential decision problems using inference algorithms (No. TR-2007-04). University of British Columbia, Computer Science.
http://mlg.eng.cam.ac.uk/hoffmanm/papers/hoffman:2007a.ps

Hoffman, M. W., Kueck, H., de Freitas, N., & Doucet, A. (2009). New inference strategies for solving Markov decision processes using reversible jump MCMC. In Uncertainty in Artificial Intelligence (pp. 223–231).

Hoffman, M. W., de Freitas, N., Doucet, A., & Peters, J. (2009). An Expectation Maximization algorithm for continuous Markov Decision Processes with arbitrary reward. In the International Conference on Artificial Intelligence and Statistics (pp. 232–239).

Hoffman, M. W., Doucet, A., de Freitas, N., & Jasra, A. (2007). Bayesian policy learning with trans-dimensional MCMC. In Neural Information Processing Systems (pp. 665–672).

Hoffman, M. W., & de Freitas, N. (2012). Inference strategies for solving semi-Markov decision processes. In L. E. Sucar, E. F. Morales, & J. Hoey (Eds.), Decision Theory Models for Applications in Artificial Intelligence: Concepts and Solutions. IGI Global.

Kueck, H., Hoffman, M. W., Doucet, A., & de Freitas, N. (2009). Inference and Learning for Active Sensing, Experimental Design and Control. In Proceedings of the Iberian Conference on Pattern Recognition and Image Analysis (pp. 1–10).

For code and pdf links, please go to http://mlg.eng.cam.ac.uk/hoffmanm/papers/
I hope you find this useful. Best.

---

### Meta-Review · Area_Chair1 · 2018-12-14
**Promising approach and text should be clarified on some points of active discussion**

**Confidence:** 3
**Recommendation:** Accept (Poster)

**Metareview:**

This paper presents a new approach for posing control as inference that leverages Sequential Monte Carlo and Bayesian smoothing. There is significant interest from the reviewers into this method, and also an active discussion about this paper, particularly with respect to the optimism bias issue. The paper is borderline and the authors are encouraged to address the desired clarifications and changes from the reviewers.